# Heat Source Models in Numerical Simulations of Laser Welding

**DOI:** 10.3390/ma13112653

**Published:** 2020-06-10

**Authors:** Tomasz Kik

**Affiliations:** Department of Welding Engineering, Silesian University of Technology, Konarskiego 18A, 44-100 Gliwice, Poland; tomasz.kik@polsl.pl; Tel.: +48-32-237-1681

**Keywords:** finite element method (FEM), welding, laser, heat source model, keyhole welding, thermal analysis

## Abstract

The article presents new possibilities for modifying heat source models in numerical simulations of laser welding processes conducted using VisualWeld (SYSWELD) software. Due to the different power distributions and shapes of a laser beams, it was necessary to propose a modification of heat source models and methods of defining the heat introduced into a welded material in the case of simulations of welding processes using solid-state and high-power diode lasers. A solution was proposed in the form of modification of predefined heat source models in the case of simulations of welding processes using solid-state disc lasers and high-power diode lasers (HPDL). Based on the results of metallographic tests and the acquisition of thermal cycles of real laser welding processes, the process of calibration and validation of the proposed models of heat sources depending on the type of device used as well as the obtained shapes of fusion beads was carried out. The purpose and assumptions of this approach towards creating heat sources were also reported, comparing exemplary stresses and cumulative plastic strain distributions for the calculation variant using a standard and modified heat source model.

## 1. Introduction

The basics of lasers were presented in 1917 by Albert Einstein in his theory of stimulated emission, where it was stated that stimulated radiation and stimulating radiation have the same properties, i.e., the same direction of propagation and the same phase of vibration and polarization. These properties therefore indicate the fact that with a beam of laser light, a concentrated portion of energy can be transported. However, a long time passed before this could be applied at an industrial scale. This was mainly due to the problem of building devices that would be able to provide adequate power so that the laser beam focused on the surface of the material would be able to heat it to a temperature suitable for the selected technological processes. At present, as this possibility now exists, the undoubted advantages of concentrated, monochromatic light combine the precision of the process with product quality and efficiency inaccessible to traditional arc welding methods. Joining material technologies using a laser beam are characterized by smaller deformations and the effect of heat on the joined elements. The use of laser devices has caused many technologies to change and has expanded the areas of application, offering an up-to-now unattainable level of product quality. With their advantages, laser technologies somehow meet ever-increasing quality requirements and are also able to address the increasing complexity of the welding technologies used [1,2].

Depending on the internal design of the device and the principle of operation, it is an option to use devices with different radiation intensity distributions across the laser beam cross-section [2]. The power density of the device depends mainly on the energy distribution in the focus of the laser beam. The shape of the electromagnetic field generated on the beam cross-section is referred to as transverse electromagnetic mode (TEM). The multitude of energy distributions that it is possible to obtain on the surface of the laser beam and in devices used in the industry, on the one hand, introduces incredible flexibility in adapting these devices to a specific application. On the other hand, it allows for many devices with significantly different properties despite, e.g., the similar maximum laser beam power offered. The existing variety of types of devices as well as beam focus shapes, radiation wavelengths, and power distributions on the laser beam focus surface means that the modern engineer has considerable possibilities for controlling the dimensions of the beads and the penetration shape obtained in processes using laser devices. However, this also causes a problem in choosing the right device for a particular application, especially taking into account the constantly shortening response time required by the market. The degree of complexity of current, modern welding technologies based on this type of equipment, along with the mentioned advantages and requirements, forces engineers and constructors to take a different approach to the design stage and search for modern tools supporting the preparation of welding and heat treatment technologies themselves [2,3,4,5,6,7].

One modern tool supporting their work is (among other programs for numerical simulation of manufacturing processes) based on the finite element method (FEM). The idea of supporting calculations was created quite a long time ago. The first attempts at prediction of residual stresses and deformations distributions were presented in 1938 by Rodger and Fletcher [8]. However, the insufficient computing power at that time blocked the use and development of these techniques for years. In 1970, Brust et al. presented the first welding numerical simulations using the finite element method [9,10,11]. Over the past several years, mainly due to the development of computers and the significant increase in available computing power, there has also been a huge improvement in computational techniques in the field of welding and heat treatment simulation. New computational techniques and a method for the mathematical description of the heat introduced into the welded material were developed. This resulted in a significant extension of the possibility of using modern computational programs in the everyday engineering practice of designers and welding engineers in industry [12,13,14,15,16,17].

Currently, numerical analyses of welding processes are usually used to optimize and validate these processes both from a technological and economic point of view, but in some cases they are used for the modification of the welded structure itself. Thanks to the continuous development of this field of knowledge, it is now possible to conduct local welding analyses, and then based on the received heterogeneous distributions of structures and material properties determine stresses and plastic deformations as well as the loads of the analyzed structure with external forces, pressures, and deformations. Based on the results obtained, it is possible to determine potential places of fatigue damage in the analyzed welded constructions. Modern software packages for numerical analyses create completely new opportunities for the modern engineer. The engineer may receive help that allows for a significant reduction in the time of production preparation or the introduction of changes in the technological process, thanks to the possibility of simulating their impact on the process result. In addition, this operation is completely safe because it takes place in the computer’s memory and does not affect the actual technological process or security. These possibilities are however also associated with a certain degree of inconvenience, which is possible to resolve. The abovementioned possibilities for conducting analyses and the amount of input data and related information make numerical simulations of welding processes difficult and complicated to carry out using FEM simulations [17].

## 2. Description of the Problem

### 2.1. Basics of Formulating the Problem of Temperature Distribution Calculations in Numerical Simulations

Before FEM computer simulations of welding processes came into general use, the only way to present temperature fields during welding was through analytical methods. However, the emergence of modern technologies, new materials, and the increasing complexity of the designed structures have made the application of analytical methods very difficult or even impossible in some cases. The development of packages for computer simulation of welding processes based on the finite element method has opened up new possibilities in the field of design support for these processes. Since numerical simulations of temperature distributions are calculated on the basis of Fourier’s differential formula, to perform calculations it is necessary to obtain the values of the heat conduction coefficient, specific heat, and density depending on the temperature [18,19,20]:(1)$$\frac{\partial T}{\partial t}=\frac{\lambda }{c\rho }\left(\frac{\partial^{2}T}{\partial x^{2}}+\frac{\partial^{2}T}{\partial y^{2}}+\frac{\partial^{2}T}{\partial z^{2}}\right)=a\nabla^{2}T$$
where *T* is the temperature, *t* is time, *x, y,* and *z* refer to the point coordinates, *a* is the thermal diffusivity coefficient, *λ* is the heat conductivity coefficient, *c* is specific heat, and *ρ* is mass density.

In the analyses presented later in the article, the VisualWeld 15.5 (SYSWELD) package of the ESI Group (Paris, France), was used, where the coupled thermo-metallurgical analysis is described by a modified heat convection equation [19,20]:(2)$$\left(\underset{i}{\sum }P_{i}{\left(\rho C\right)}_{i}\right)\frac{\partial T}{\partial t}-\nabla \left(\left(\underset{i}{\sum }P_{i}\lambda_{i}\right)\nabla T\right)+\sum_{i<j}L_{ij}\left(T\right)\cdot A_{ij}=Q$$
where *P_i,j_* refers to the proportions of phases with indexes *i* and *j*, respectively, *L_ij_(T)* is the latent heat of *i* to *j* transformation, *A_ij_* is the proportion of phase *i* transformed to *j* in a time unit, and *Q* is the total amount of introduced heat.

The consequence of using Fourier’s differential equation is the non-linearity of the calculated temperature field distributions. It is also necessary to consider that the temperature field distributions are calculated based on the mathematical description of a heat source model represented by thermal flow density into material. In many cases it is a moving heat source and the coordinates describing its location are related not only to the space coordinates but also to the duration of the process. The formation and development of welding strains and stresses is associated with several very closely related factors such as clamping conditions, thermo-mechanical material properties, type of used welding technology, process parameters, ambient temperature, and cooling conditions, among others [14,15,16,17,18,19,21,22].

### 2.2. History of Heat Source Models in Numerical Simulations

Models of heat sources used in numerical simulations of welding processes can be generally divided into two groups:—Models with a homogeneous distribution;—Models with heterogeneous distribution [23]

In 1941, Daniel Rosenthal was the first person to apply Fourier’s law to a moving heat source [24]. This resulted in the first model of a heat source with a homogeneous distribution. Unfortunately, the biggest disadvantage of this solution was the fact that all the energy of the process was concentrated in one point, which in effect meant that the use of this model did not in any way allow for the determination of the shape of the molten pool and the depth of penetration. This solution was then modified, but an acceptable compliance of temperature distribution prediction was obtained only at a distance from the heat source itself. A certain extension was the linear model of the heat source proposed in 1969 by Pavelic [25]. Thanks to this solution, an attempt was made to determine the temperature distribution in a two-dimensional element. Still, a problem was the inability to precisely map the temperature gradient over the thickness of the model. A few years later, Friedmann proposed a heat source model that had a volume distribution. Equations describing cylindrical and square models of the heat source were successively developed. Thanks to these activities, the results obtained became closer and closer to those obtained from real tests. It should be noted that at that time processes with high power density were already being described using rectangular (electron beam) and cylindrical heat source models (laser beam). Nevertheless, the degree of simplification of these models did not ensure a complete mapping of the nature of these processes [23,26]. In 1983, Eagar and Tsai introduced the first model with the heterogeneous distribution. The proposed model to some extent allowed for the weld pool geometry to be shown by using the Gaussian heat distribution on the surface [27]. On this basis, a disk model was later created. Its use was still somewhat limited due to the inability to determine the depth of operation of the model [23].

Over the years, in association with the development of numerical analyses of welding processes, many mathematical descriptions have been created regarding models of heat sources. These range from the simpler point models that limit all energy of the welding process to one point (mesh node), through to linear and circular models, up to more advanced forms. However, it was in the early 1980s that John Goldak proposed a widely used model in the shape of a double ellipsoid. Although it is currently mainly used for simulation of arc welding processes, i.e., a technique with a liquid metal pool, after appropriate modifications it can also be used to simulate selected laser welding processes, as shown later in the article [16,21,22,23,28,29].

### 2.3. Typical Models of Heat Sources Used in Numerical Simulations

In commercially available software dedicated to solving welding process problems, three types of predefined heat source models are usually used:—The Gaussian surface heat source model;—The Double ellipsoid heat source model (Goldak’s model);—The conical heat source model

The Gaussian distribution model, in which energy is distributed according to the course of the Gaussian curve, is usually used in simulations of welding processes with high power densities, i.e., laser, plasma, and microplasma or electron beam welding (Figure 1). With the help of this heat source model it is possible to map deep penetration while maintaining a small width, characteristic of “keyhole” welding techniques. In the VisualWeld (SYSWELD) environment, a normal distribution heat source is a predefined and recommended model used in simulations of a surface heat treatment processes, including laser beam hardening [20]. The Gaussian surface heat source model is described as in [20]:(3)$$Q\left(x,y,z\right)=Q_{0}exp\left(-\frac{x^{2}+y^{2}}{r_{e}^{2}}\right)$$
where *Q_0_* is the amount of heat introduced by the source, and *x, y,* and *r_e_* are the geometrical parameters describing the heat source position.

The most popular heat source model currently used in commercial programs for numerical analysis of welding processes is the volumetric double-ellipsoid heat source model, also known as Goldak’s model. This model is built of two ellipsoids placed perpendicular to each other, allowing the size of the source to be determined in the plane resulting from the position of the source perpendicular to the direction of welding (Figure 2).

Unlike other volumetric models used for arc welding simulations, Goldak’s model is described by two equations individually for each part of the ellipsoid separately [17,18,20,21]. For the front part of the ellipsoid, the equation is:(4)$$Q_{f}\left(x,y,z\right)=\frac{6\sqrt{3}f_{f}Q}{abC_{f}\pi \sqrt{\pi }}exp\left(\frac{-kx^{2}}{a^{2}}\right)exp\left(\frac{-ly^{2}}{b^{2}}\right)exp\left(\frac{-mz^{2}}{c^{2}}\right)$$
and for the rear part of the ellipsoid, the equation is:(5)$$Q_{r}\left(x,y,z\right)=\frac{6\sqrt{3}f_{r}Q}{abC_{r}\pi \sqrt{\pi }}exp\left(\frac{-kx^{2}}{a^{2}}\right)exp\left(\frac{-ly^{2}}{b^{2}}\right)exp\left(\frac{-mz^{2}}{c^{2}}\right)$$
where *Q_f_, Q_r_* are the volumetric heat flux density in front and rear part of the model, respectively, (W/m^3^), *Q* is the total introduced power, and *a b, c_f_,* and *c_r_* are respectively the width, depth, and length of the front and the rear part of the estimated molten pool.

The coefficient of heat transfer efficiency to the welded material is determined depending on the welding method being analyzed [17,20].

How much of the total heat is transferred to the front and back of the ellipsoid is determined by the parameters *f_f_* and *f_r_*. These are constants that affect the distribution of energy flow to the material, which can somewhat compensate, e.g., the tilting effect of the welding torch. As practice has shown, the relation of *f_f_* to *f_r_* is usually accepted as a ratio of 60:40. It is important to remember that these parameters must satisfy the dependence *f_f_ + f_r_* = 2 [17,30]. Determining the values of coefficients *k*, *l*, and *m* is a much more complex issue. When using an unmodified heat source model, all three factors assume a value of 3. An example of such an application can be e.g., simulation of the Metal Manual Arc (MMA) welding process. However, in the case of simulation, e.g., of the Gas Metal Arc (GMA) welding process, the values of these coefficients often must be modified experimentally to obtain the correct shape of the molten pool. Currently, this model so far best describes the actual state of arc welding methods [16,17,18,20,21,22].

In analyses of welding processes characterized by high energy densities, i.e., laser, electron beam welding or plasma welding, a conical source model with a normal distribution is usually used (Figure 3).

The mathematical description of volumetric heat flow density into the material is described as follows [17,20]:(6)$$Q\left(x,y,z\right)=Q_{0}exp\left(-\frac{x^{2}+y^{2}}{r_{0}^{2}\left(z\right)}\right)$$
(7)$$r_{0}\left(z\right)=r_{e}+\frac{r_{i}-r_{e}}{z_{i}-z_{e}}\left(z-z_{e}\right)$$
where *Q_0_* is the maximum value of volumetric heat flux density, *r_e_,* and *r_i_* are respectively the upper and lower cone radius dimensions parameters, *z_e_,* and *z_i_* are the cone length parameters, and *x, y,* and *z* are point coordinates (Figure 3).

The use of a moving heat source model in numerical simulations of welding processes requires the use of an appropriate calibration procedure. Only in this case is it possible to correctly determine the stresses and deformation levels and distributions, calculated based on previously obtained results of temperature fields and individual metallurgical phase distributions. Incorrectly performed calibration of the heat source model usually leads to significant differences in the results obtained as compared to the reality. The use of predefined models does not mean, however, that the user cannot create a combination of these sources to obtain e.g., a set suitable for the simulation of hybrid welding consisting of a conical model for the laser beam and the double-ellipsoid model for mapping the effect of arc welding [31]. It is possible to use self-defined heat source models as well as other computational techniques that do not use moving heat source models. Unfortunately, they often require initial calibration based on the analyses performed using them [17].

### 2.4. The Shapes of Beads Geometry Obtained in the Case of Laser Welding Processes—Formulation of the Research Problem

As mentioned before, the conical heat source model is the most commonly used heat source model in the case of the “keyhole” welding technique. However, the model does not always allow for correct results as compared to the reality. While it works relatively well when calculating a process partial penetration, when there is full penetration obtained it is not possible to model the bead expansion characteristic of the laser welding process in the root area when welding thicker parts. In the laser welding process, essentially three different bead geometry shapes can be obtained. In the case of welding without full penetration, the shape is usually narrow, cylindrical, or slightly expanded upwards—the so-called bell shape (Figure 4a).

The situation changes after achieving full penetration of the bead. In this case, basically two shapes of the melted metal area are obtainable. One has a wide face area and slight widening in the area of the root (Figure 4b). The situation changes after achieving full remelting. In this case, two shapes of the melted metal area are obtainable. The first is due to wide melting in the face area and slight expansion in the ridge area. The second, in turn, shows wide remelting in both the face and root areas, with a narrow area melted in the middle of the bead (hourglass shape) (Figure 4c). The bead shape obtained when welding with full penetration depends on the type of device used, the process parameters, and sheet thickness as well as the type of material being welded [29,31,32,33,34,35]. The information found in the literature shows that the nature of the bead geometry changes at full penetration results from several complex phenomena that determine the flow of liquid metal in the molten pool and the convective heat that affects the shape of the bead [29,34,35,36]. A complete solution to this type of problem requires numerical simulations using complex fluid mechanics models. Unfortunately, the aforementioned simple models used in commercial software for simulation of welding processes are models based on thermal conductivity, where this flow is not taken into account. Several studies have appeared in the literature on attempts to solve this problem. In 2017, Flint proposed a model that is an extension of the double-ellipsoid model that can be used for narrow-groove and keyhole weld configurations [37]. Wu et al. also proposed some modifications of the conical model where a modified three-dimensional cone model (TDC) was used to simulate the plasma welding process with the mesh technique [38]. Similarly proposed by Farrokhi et al. in the case of hybrid welding, combined models based on a double three-dimensional conical heat source model for the reconstruction of “keyhole” welding conditions with partial and full penetration indicate the fact that in order to obtain the correct shape of the melted area, it is necessary to combine two or more volumetric models of heat sources [29]. In 2018, for the investigation of a dual-beam laser welding of aluminum sheets, He et al. proposed modification of the cone-shaped heat source model for the lap joint configuration. Based on the proposed model and the experimental validation procedure, investigations of hot cracking were conducted [39].

From a scientific point of view, this is the most correct reasoning aimed at precisely and accurately reproducing the obtained bead shape in the heat source model. For a typical commercial user of mathematical notation of this type, proposed models of heat sources can be a real obstacle to everyday use. Not every user has the programming knowledge needed to create their own heat source model description. Therefore, research should be continued to propose the simplest and most effective way to reproduce the actual geometry of the bead during numerical simulations of welding processes.

## 3. Modification of Predefined Heat Source Models in the Case of Laser Bead-On-Plate Welding

### 3.1. Test Stand and Preliminary Attempts to Use Predefined Heat Source Models

To propose a solution that would be simple to use for a typical commercial user, welding tests were carried out on two diametrically opposed laser devices. The first test stand was equipped with a Trumpf TruDisk 3302 (Ditzingen, Germany) disk laser with a maximum beam power of 3300 W (Figure 5a). This laser allowed us to obtain a high quality beam with a wavelength of 1030 nm while maintaining a beam spot diameter of 200 µm. The laser head was equipped with a 200-mm collimator lens and a 200-mm focusing lens. The beam parameter product (BPP), as claimed by the manufacturer of this laser, was <8.0 mm × mrad.

The second type of laser used in the research was a high-power diode laser (HPDL) (Figure 5b). The ROFIN DL 020 direct diode laser from ROFIN Sinar (Hamburg, Germany) allows for the following rectangular beam spot dimensions to be obtained: 1.8 mm × 6.8 mm at 82 mm focal length or 1.8 mm × 3.8 mm at 32 mm focal length (with an additional focusing lens). The maximum output power of the HPDL ROFIN DL 020 is 2200 W with the wavelength at 808 nm, with very even energy distribution across the beam spot (multimode TEM).

For preliminary tests using a disk laser, bead-on-plate welding tests were performed on a 6.0-mm-thick sample of S355 steel at 1500 W with a welding speed of 8.3 mm/s, and on a 4.0-mm sample of AISI304 steel (X5CrNi18-10) at 1000 and 1500 W and 8.3 mm/s welding speed (Table 1 and Table 2).

To simulate bead-on-plate welding tests, a three-dimensional numerical model of a 6.0-mm-thick flat sheet was prepared. To ensure the correct mapping of heat dissipation conditions, a boundary condition was adopted describing heat exchange with the environment on all external surfaces of the model. Boundary conditions for clamping were adopted as standard conditions corresponding to welding of the sample placed without additional fastening on the laser station table. The calculations were performed in the “heat source input fitting” module as a “steady state”-type analysis (analysis carried out at only one moment of time after the process was established), using a normal distribution heat source model (Gaussian surface heat source model) with the following parameters: laser beam power, 1500 W; welding speed, 8.33 mm/s; *r_0_* = 2.0 mm (Figure 1). As a result of tests carried out at the laser stand, a bead width of 2.08 mm and a maximum penetration depth of 5.20 mm were obtained. For comparison, the results of numerical simulation allowed the determination of these dimensions at the level: bead width 2.02 mm and maximum penetration depth 5.31 mm (Figure 6). In the case of the parameters used, a similar molten pool shape was obtained in the numerical simulation. The tests carried out showed that as the laser beam power (and penetration) increased, the compatibility of the numerical simulation with the reality for this heat source model decreased significantly.

In the case of AISI304 steel samples, a three-dimensional numerical model of a flat sheet with a thickness of 4.0 mm was made analogously. The boundary conditions were also set to simulate no sample fixing. The calculations were performed using a conical source heat source model using the following parameters of the heat source model (Figure 3):

DL1.0_8.33—Laser beam power: 1000 W; welding speed: 8.33 mm/s; r_e_ = 2.44 mm; r_i_ = 1.12 mm; z_e_ = 0; z_i_ = 3.16 mm

DL1.5_8.33—Laser beam power: 1500 W; welding speed; 8.33 mm/s; r_e_ = 3.1 mm; r_i_ = 2.4 mm; z_e_ = 0; z_i_ = 4.0 mm.

As a result of the tests, beads with the aforementioned shapes were obtained, with bell and hourglass shapes with the geometrical dimensions presented in Table 2 and Figure 7. While there is considerable agreement between the dimensions measured on real samples and the same dimensions obtained from the results of numerical analyses carried out using a conical model of the heat source, the shape of the real and calculated beads varied significantly (Figure 7).

In summary, the modeling tests carried out using implemented models of heat sources, i.e., the Gaussian surface and conical heat source model, showed that there is a need to develop modifications to obtain the actual shape of calculated beads.

### 3.2. Assumptions and Construction of a Modified Numerical Model in the Case of a Disk Laser

It is possible to create a combined model consisting for example of two conical sources or one conical and other cylindrical sources (Figure 8). However, in this case it is problematic to divide the amount of energy for each of these parts (as shown by the author’s experience, not from the volume ratio of both models). This is due to different heat distribution conditions in individual batches of materials as well as heat dissipation conditions. Additionally, this requires defining two trajectories along which the heat source model moves (Figure 8). Nonetheless, it should be emphasized that this simulation method allows for a significant improvement in the results obtained and a better mapping of the fusion line shape.

Considering the above, the simplest methods for creating a model with a high correlation coefficient of results with reality are still being sought as simply as possible. To solve this problem, it is possible to use the functionality offered by the VisualWeld 15.5 package from the ESI Group, France. In this software, it is possible to specify elements in the model mesh that will be only affected by a standard heat source model. Simply put, only the selected mesh elements covered by the standard heat source model will be heated. Using this function, it is possible to modify the shape of the fusion line in a relatively convenient and simple way (Figure 9).

To use this method, however, it must be assumed that in principle, it applies to all numerical analyses of welding processes. The shape of the bead fusion line (based on the macroscopic metallographic picture) should be treated as input data, not the result of the analysis. The task of numerical analysis performed based on the heat conduction equation, not the problem of fluid mechanics, is to obtain additional information on the distribution of metallurgical phases, hardness, stresses, or joint deformations during and after the process. The lack of consideration of complicated mathematical models describing the convective movements of liquid metal in the weld pool itself means that the shape of the fusion line, which is highly influenced, must be set by the user. This is justified by the fact that in simulations with available programs for numerical analyses of welding and heat treatment processes many factors are ignored that in welding processes have a significant impact on the shape of the fusion line. This even includes the impact of the shielding gas used or the method of transferring metal in the arc during GMA welding, which are factors that have a huge impact on the nature and intensity of the abovementioned movements of liquid metal in the pool. Thus, the shape of the fusion line is a very important input parameter at the calibration stage of the calculation model. Following the proposed assumption, the calibration of the heat source model is not only about entering the source power and dimensions of the heat source model corresponding to actual or expected values, but also manipulation of the heat source model shape and the area it affects (red-colored area in Figure 9 marked as “LOAD”) so that the obtained result is as close as possible to the result of metallographic tests of the bead used in model calibration. The user, by appropriately selecting the parameters of the heat source model and the mesh elements that he exposes, has much greater possibilities of mapping the actual shape of the melted area.

### 3.3. Calibration and Validation of the Model with a Modified Method of Heat Input in the Disk Laser Bead-On-Plate Welding Process

To confirm the assumptions of the proposed simulation method, a TruDisk 3302 disk laser bead-on-plate welding process on 6.0-mm-thick AISI304 steel samples was carried out (Table 3). Data from metallographic tests of the obtained fusion beads were used to calibrate the heat source model. To validate the calibrated model, 4 samples with a thickness of 4.0 mm from AISI304 steel were also analyzed, to which the bottom surface was welded with three K-type thermocouples (Ni–NiCr wire with diameter of 0.2 mm) to record the heat cycles of the welding process. Thermocouples with accuracy ±0.0075 × T in a temperature range from −40 to +1200 °C (according to EN 60584-1 standard) were connected via compensation cables with an Agilent 34970A recorder by Agilent Technologies Inc., Santa Clara, CA, USA [42]. The arrangement scheme was as follows: the TC3 thermocouple was placed on the bead axis in the middle of its length, the TC2 thermocouple was placed 5.0 mm from the bead axis in the middle of its length, and the TC1 thermocouple was also placed in the bead axis at 5.0 mm from TC3 (Figure 10).

Then the symmetrical three-dimensional discrete model, containing 28,080 3D elements and 30,622 nodes for 6.0-mm-thick samples and 21,518 3D elements (6 node linear wedges and 8 node linear bricks) and 23,706 nodes for 4.0-mm-thick samples was prepared. Mesh was refined in the weld and heat-affected zone (HAZ). The applied boundary conditions simulating sample attachment corresponded to its free placement on the laser table throughout the whole process of welding and then its cooling. In addition, a boundary condition of symmetry was added to the surface of the model division (Figure 11). The boundary condition describing heat dissipation to the environment was implemented by defining convective heat dissipation to the environment at 20 °C and radiation on all external surfaces of the model. As a heat source model, proposed as volumetric, a conical model with a modified “LOAD” area was used (Figure 11). The calculations to define the material database contained the thermal, mechanical, and metallurgical material properties, which were dependent on temperature and metallurgical phase proportions.

Based on the results of metallographic tests, the calculation models were calibrated, and the final results of numerical analyses were compared with the corresponding shapes of the actual beads obtained in the welding tests (Figure 12, Figure 13 and Figure 14). On this basis, a comparison was also made of the measured and calculated basic geometric dimensions of the beads obtained (Table 4).

Following the assumptions of the proposed model preparation method, re-calculations were also performed for the DL1.5_8.33 sample analyzed in sub-chapter 3.1 (Figure 7). The results of the repeated numerical analysis showed that there was a significant improvement in the quality of the fusion line mapping using the same model of a conical heat source but with a modified area of the loaded elements of the mesh (previously all mesh elements covered by the cone were loaded), (Figure 7 and Figure 15).

TC_1-TC_4 samples were prepared to collect information for model validation using thermal cycle runs. After calibrating the calculation model (Figure 16), the registered and calculated points corresponding to the location of thermocouples were also compared, as was the course of the process thermal cycles (Figure 17, Table 3).

To show the purpose of such precise calibration of the model and reproduction of the actual fusion line shape, for selected DL_T3 beads the von Mises reduced stresses and cumulative plastic strain distribution were also calculated for the simulation case with a standard heat source model and unmodified and modified load element area (Figure 18 and Figure 19).

### 3.4. Assumptions and Construction of a Modified Numerical Model in the Case of a High-Power Diode Laser

The second laser device analysed for numerical simulations in the presented research was a high-power diode laser. This source is an interesting device that is noteworthy from a numerical simulations point of view as due to its internal structure (laser diode packages in the form of laser rods) it emits a rectangular laser beam, with beam spot dimensions of 1.8 mm × 6.8 mm at a focal length of 82 mm or 1.8 mm × 3.8 mm using an additional focusing lens and focal length of 32 mm. Due to the multimode energy distribution on the surface of the focus of the laser beam, the value of this energy is almost constant on the entire surface (Figure 20). 

This type of distribution caused the power density on the surface of a relatively large focus of the laser beam to be much lower than on the previously used disk laser (maximum 3.2 × 10^4^ W/cm^2^ as compared to 1.05 × 10^7^ W/cm^2^). The inability to conduct the welding process with the “keyhole” technique in the case of these lasers is compensated by their advantages when used in surface treatment and surfacing processes (wide beam with uniform energy distribution) [43].

The shape of the laser beam focus and the even distribution of energy of the laser radiation on the surface of this focus also require a special approach in the case of numerical analyses due to the fact that models that are usually not well represented are not available. While the historical rectangular heat source model described here corresponds to the beam shape, as already mentioned the user of modern programs dedicated to welding and heat treatment processes usually has three predefined models of heat sources: the Gaussian surface model, the double-ellipsoid model, and the conical model. As before, in this case it is possible to load the model’s heat source with selected elements of the model’s mesh and utilize its dimensions to control the width and depth of bead fusion.

Taking into account the internal structure of this type of laser (single diode emitters with single-mode distribution), the first approximation of this heat source can be a model in the form of two rows (due to the longitudinal dimension of the laser beam focus of 1.8 mm) of individual normal (Gaussian) distributions moving along the trajectory placed in the load area, specifying the bead obtained with some approximation (Figure 21a). This brings with it the aforementioned disadvantage associated with the need to define a large number of trajectories and heat source models as well as to divide the total amount of heat introduced by the appropriate number of sources used. Such an action does not reflect the construction of the laser rod in such a way where there are emitters with a power of 1–2 W. Therefore, with a laser power of 2000 W the grid of the discussed model and the number of heat sources necessary to define the model would cause the model to cease to be real due to the quantity of grid elements used in the numerical model (hundreds or even thousands of small heat source models corresponding to individual diode emitters). The tests carried out showed that even with a much rarer mesh of this type (and a reduced number of used heat sources), the solution ensured high compatibility of the shapes of the calculated beads with those obtained as a result of real tests (Figure 22, Table 5). The aforementioned much lower power density on the surface of the laser beam focus and the use of processes with the “weld pool” technique (conduction mode) in a similar way to arc welding inspired an attempt to adapt the most popular model, which is the double-ellipsoid model. Due to the ease of use in modeling the “weld pool” technique, the need to define only one trajectory of heat source model movement, and the ability to determine the depth of model interaction, attempts were made to confirm the usefulness of this solution. For this purpose, a model was built in which, as in the previous case, the “LOAD” area roughly corresponded to the dimensions of the obtained bead, while the double-ellipsoid heat source model was much wider than this area to ensure uniform energy distribution on the surface of the loaded area (Figure 21b). Other dimensions of the heat source model are related to the depth of penetration of the bead and the longitudinal dimension of the laser beam focus.

To verify the assumptions of the presented model, a three-dimensional discrete model in VisualWeld (SYSWELD) package was created, containing 11,120 3D elements and 13,243 nodes. The boundary conditions used, simulating the clamping of the sample, corresponded to its free placement on the table of the laser stand throughout the welding process and then its cooling. The boundary condition regarding heat dissipation was met by convective dissipation to the environment at 20 °C and radiation was defined on all external surfaces of the model. Numerical analyses were carried out for a model consisting of 12 pieces of the Gaussian surface heat source models (in two rows placed behind each, 6 models in each) and 6 trajectories determining the movement of these sets, located at a distance of 1.0 mm from each other. The second model was a model consisting of 22 Gaussian surface heat source models (in two rows placed behind each with 11 models in each) and 11 trajectories determining their movement at a distance of 0.5 mm from each other (Figure 21a). In the case of the described model, built using a single double-ellipsoid heat source model, analyses of 6 variants differing in the dimensions of the heat source model itself were performed (Figure 22, Table 5). In each of the analyzed cases the values of laser beam power and welding velocity were the same and were respectively 1400 W and 2.5 mm/s (Table 5).

### 3.5. Calibration and Validation of the Model with a Modified Method of Heat Input in the High-Power Diode Laser Bead-On-Plate Welding Process

To verify the proposed model, numerical analyses were performed using the model of assumptions proposed earlier, and a three-dimensional discrete model was created containing 38,320 3D elements and 41,148 nodes. The applied boundary conditions simulating sample clamping corresponded to its free placement on the laser table throughout the whole process of welding and then its cooling. The boundary condition regarding heat dissipation was met by convective dissipation to the environment at 20 °C and radiation defined for all external surfaces of the model. As a heat source model, a double-ellipsoid model was used which acted on selected elements of the model’s mesh (Figure 23). In this case, the calculations using the defined material database containing the thermal, mechanical, and metallurgical material properties depended on temperature and metallurgical phase proportions.

For the calibration of the proposed model, the results of the bead-on-plate high-power diode laser welding process on 10-mm X37CrMoV5-1 tool steel samples were used (Table 6 and Table 7). Based on the results of metallographic tests, the calculation models were calibrated and the final results of numerical analyses were compared with the corresponding shapes of actual beads obtained in the real tests (Figure 24). On this basis, a comparison was also made of the measured and calculated basic geometric dimensions of the beads obtained (Table 7).

To validate the model, similarly to the case of disk laser welding simulations the recorded thermal cycles were used. K-type thermocouples (Ni–NiCr wires with a diameter of 0.2 mm) were welded in the holes made at the bottom of the samples so that the measuring points were located at 0.5, 1.0, 2.0, and 3.0 mm from the welded sample surface (Figure 25). Thermocouples with accuracy ±0.0075 × t in a temperature range from −40 to +1200 °C (according to PN-EN 60584-2) were connected via compensation cables with Agilent 34970A recorder. After calibrating the calculation model (Figure 24), the registered and calculated thermal cycles in the points corresponding to the location of thermocouples were also compared (Figure 26).

## 4. Conclusions

A properly prepared and conducted numerical analysis consists of three important stages: precise preparation of the material base, calibration, and model validation. A properly developed material base, created based on tests on the behavior of the material under the influence of various temperatures and thermal cycles, allows for a faithful reproduction of the behavior of materials considering its thermometallurgical and mechanical nature. The stage of building and calibrating the model is important because the variety of methods used and the effects obtained as a result do not allow for the creation of one universal model. Each process must be treated individually by the user, considering the special features of the welding heat source used and the most accurate description of the conditions of the process. Model validation allows for obtaining an appropriate degree of model reliability in further applications.

The presented results of preliminary tests using a disk laser proved the need to supplement the portfolio of implemented models of heat sources with those allowing the correct geometry of the beads in laser welding to be obtained (Figure 6 and Figure 7). Bearing in mind the mentioned desire for the greatest convenience and ease of use of models, it was proposed to extend the method of laser beam modeling or another method of welding with the “keyhole” technique (characterized by high power density). The combination of the already-present heat source model with the possibility of defining selected mesh elements which are affected by it allowed for easy and effective control of the shape of the fusion line of the obtained bead (Figure 9 and Figure 12, Figure 13, Figure 14 and Figure 15). On knowing the shape of the fusion line from real tests it was possible to calibrate the heat source model in such a way that the calculated shape of the molten pool corresponded to that from real tests. This was well-visible from the comparison of the bead cross-sections and calculated results. Interestingly, on the longitudinal cross-sections and in the corresponding results of numerical analyses, the temperature distributions corresponding to the molten pool showed considerable convergence with the obtained fusion shapes, especially when conducting the process using the “keyhole” technique. The liquid metal moved in a specific way in the gas channel formed by the beam, which was not considered in the numerical analyses based on the heat conduction equation.

The results of the validation seem to confirm that a properly calibrated model corresponds thermally to the situation recorded during actual welding tests (Figure 17). It should also be noted that in the case of thermocouple measurements, there may be disturbances in the registration of thermal cycles. In Figure 17b for the cycle recorded by TC3 it can be seen that, after recording the signal increase as a result of melting through the sample, the thermocouple located in the bead axis was damaged by liquid metal. For this reason, the corresponding course of the calculated thermal cycle was not provided. Incorrect registration of a thermal cycle with a thermocouple could also occur with incorrect positioning of the thermocouple, i.e., plane positioning or installation in drilled holes at the wrong depth, as with thermocouple TC3 in Figure 26b. These types of errors can lead to incorrect indications and in extreme situations it is impossible to register the correct thermal cycles. Therefore, in such cases it is also important that the validation is carried out based on more than one signal recorded in the actual process.

When comparing the recorded and calculated thermal cycles, there are sometimes also visible differences in the value of both the maximum value of the registered/calculated temperature as well as the course of the curves themselves (cooling speed). These differences may of course result from a simplified definition of heat exchange with the environment, convection, and emissivity coefficient values (even when they are dependent on temperature changes) [45]. This may also be due to the fact that the thermocouple tip has a certain diameter which is not taken into account when the thermal cycle is collected at a single mesh node. This is especially visible when fine mesh is used where the thermocouple tip can cover few nodes, and in case of laser welding the temperature gradients are much higher in comparison to arc welding and therefore this phenomenon is noticeable. In this situation, several nodes of the grid, differing in the maximum temperatures of the calculated thermal cycle, may be in the area covered by the thermocouple tip in the case of a real test. This should be kept in mind, especially in the case of process validation with numerical simulations, where the thermal cycles are very short and as already mentioned earlier, the temperature gradients are large so the values in the adjacent mesh nodes change very quickly.

Therefore, it is necessary to pay attention to the fact that the correct calibration of the heat source model, ensuring the convergence of the model with reality, is necessary because the heat distribution as well as the cooling speed in individual areas of the welded elements later affect metallurgical phases distribution, , grain size, and plastic deformation, and thus also stresses and strains distributions. This was confirmed by the calculated distribution of reduced stresses in the sample for the variant using the standard heat source model and the proposed extension. There are visible differences in the calculated maximum stress values as well as their distribution (Figure 18). The situation is similar in terms of cumulative plastic strain case distribution in both analyzed cases (Figure 19). It is particularly important that in the second case higher stress values were obtained, and thus a more rigorous assessment of the level of stresses and the possible danger associated with it was found.

The analyzed case of the high-power diode laser with a characteristic rectangular shape of the laser beam focus shows that more and more often the users of this type of program have to resort to the use of more complex calculation techniques to obtain correct results. The combination of knowledge on the construction of the device itself and actual welding tests carried out at the laser station allowed us to propose a simple and effective method for the simulation of an even distribution of laser radiation in this type of device (Figure 21 and Figure 22). Similar values of geometric dimensions obtained for the tested variants of the model testify to the considerable flexibility of this method and thus also guarantee its considerable versatility, as confirmed in this case by tests and calculations carried out to validate the proposed model (Figure 24 and Figure 26, Table 5). The aforementioned lower temperature recorded for TC3 in Figure 26b was caused by the incorrect position of the thermocouple in the measuring hole. As later tests showed, the thermoelement tip was welded on the wall of the hole and not on its bottom as assumed.

As a summary it should be stated, as the presented considerations have shown, that the implemented models of heat sources in commercially available programs for the numerical analyses of welding processes are not the only way to solve the analyzed problems. The possibility of introducing often simple changes or modifications to the built models allows them to be adapted in a much better way to the expected results. However, it is very important in this case that each of these modifications requires accurate calibration and validation of the proposed model based on the real experimental results. Otherwise, the results of numerical analyses may lead to erroneous conclusions.

## Figures and Tables

**Figure 1 materials-13-02653-f001:**
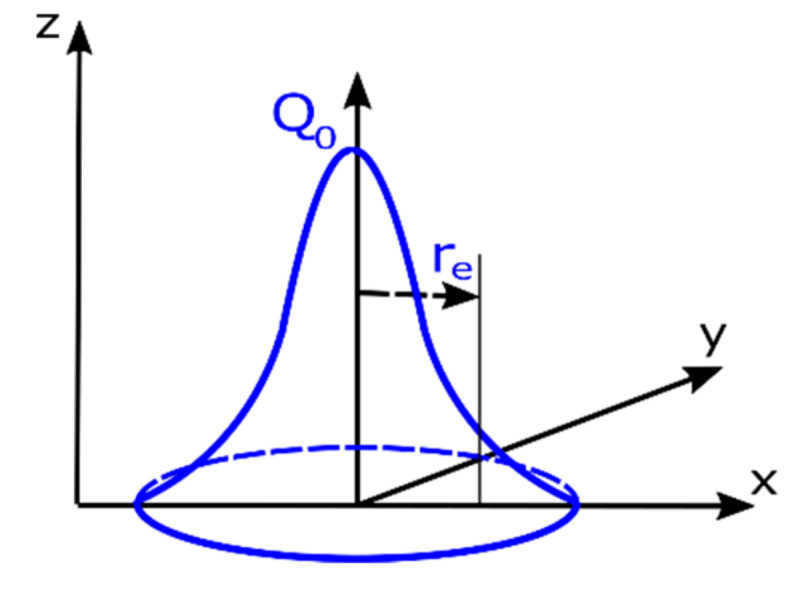
Gaussian surface heat source model [20].

**Figure 2 materials-13-02653-f002:**
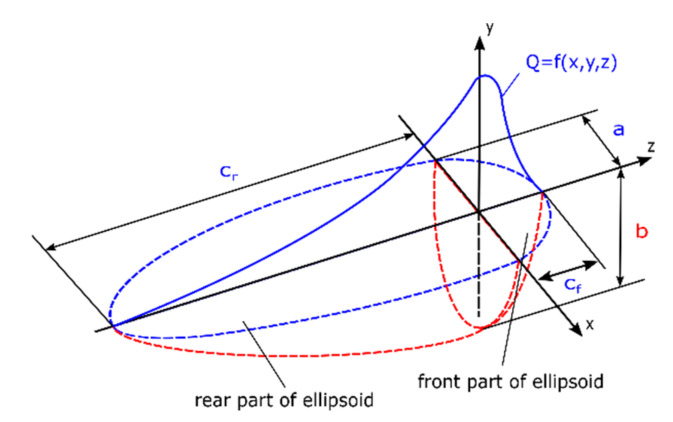
Double ellipsoid heat source models [17,18,20,21].

**Figure 3 materials-13-02653-f003:**
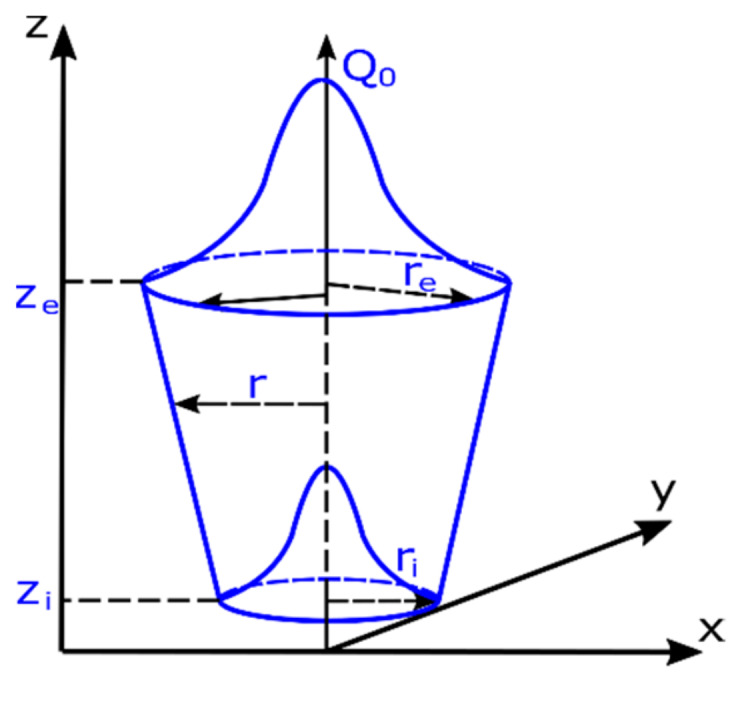
Three-dimensional conical heat source models [17,20].

**Figure 4 materials-13-02653-f004:**
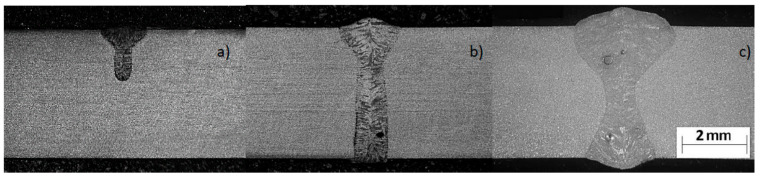
Examples of macrographs performed with disc laser welding (**a**) with partial penetration (bell shape), and (**b**) and (**c**) with full penetration (with a narrow and wide root known as the hourglass shape).

**Figure 5 materials-13-02653-f005:**
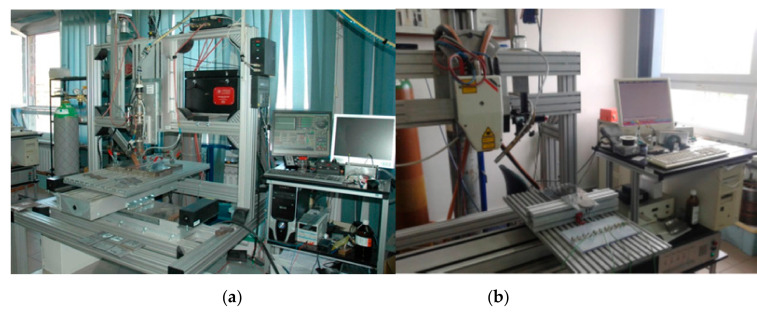
View of test stands equipped with (**a**) TruDisk 3302 disk laser with 3300 W laser beam power and (**b**) a Rofin DL020 high-power diode laser with 2200 W beam power.

**Figure 6 materials-13-02653-f006:**
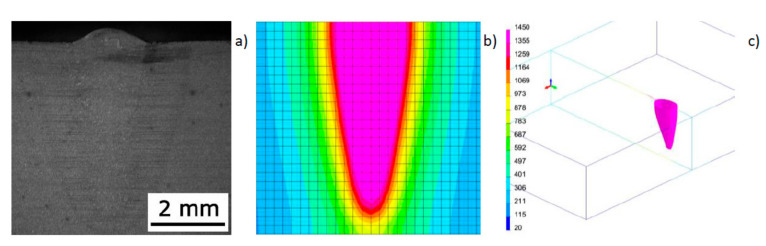
View of the macrostructure of the bead (**a**) in bead-on-plate welding tests performed on a 6.0-mm-thick sample of S355 steel using the Trumpf TruDisk 3302 disk laser at laser beam power of 1500 W and a welding speed of 8.3 mm/s; (**b**) cross-sectional view of the molten pool obtained as a result of numerical simulation; and (**c**) the 1450 °C iso-surface.

**Figure 7 materials-13-02653-f007:**
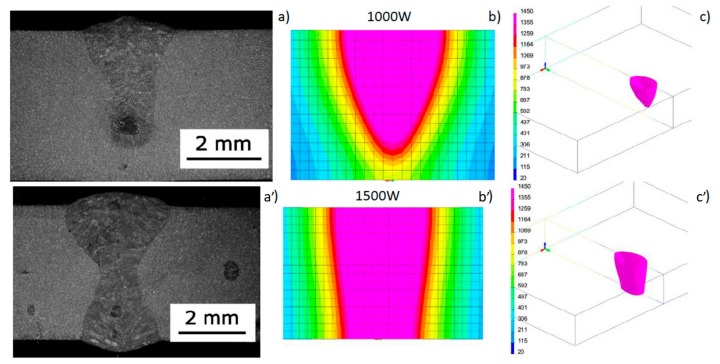
View of the macrostructure of the bead (**a**, **a’**) in bead-on-plate welding tests performed on a 4.0-mm-thick sample of AISI304 steel using the Trumpf TruDisk 3302 disk laser at laser beam power accordingly 1000 and 1500 W and welding speed of 8.3 mm/s; (**b**, **b’**) a cross-sectional view of the molten pool obtained as a result of numerical simulation; (**c**, **c’**) the 1450 °C iso-surface.

**Figure 8 materials-13-02653-f008:**
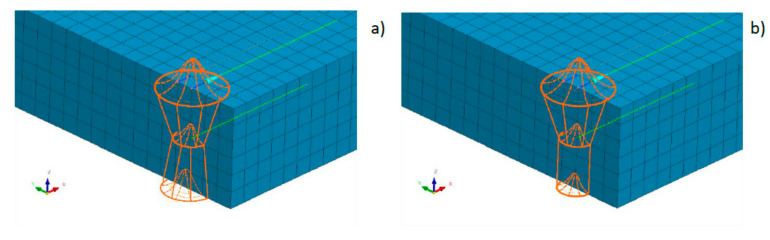
A view of heat sources models consisting of (**a**) two conical sources and (**b**) a conical and cylindrical source (green lines correspond to trajectories along which the sources move).

**Figure 9 materials-13-02653-f009:**
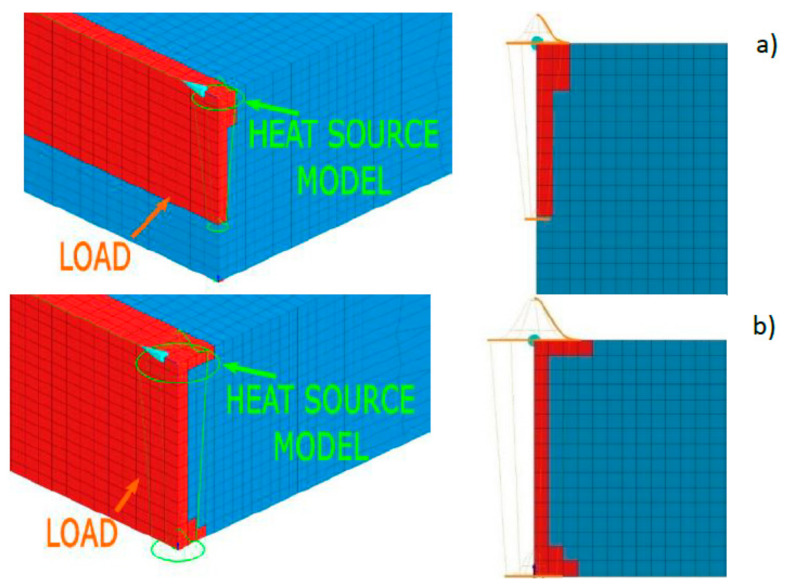
Examples defining the loaded areas of the model (network) and location of the heat source model in the case of laser welding with the mesh technique with (**a**) incomplete and (**b**) full penetration.

**Figure 10 materials-13-02653-f010:**
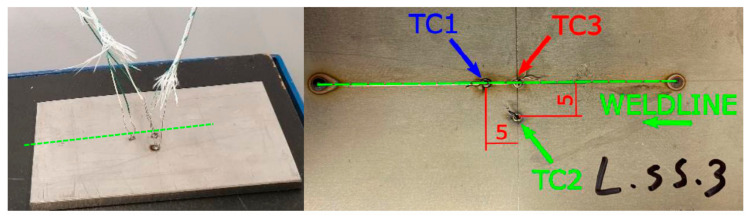
Diagram of the location of thermocouples relative to the bead axis at the bottom of the test sample.

**Figure 11 materials-13-02653-f011:**
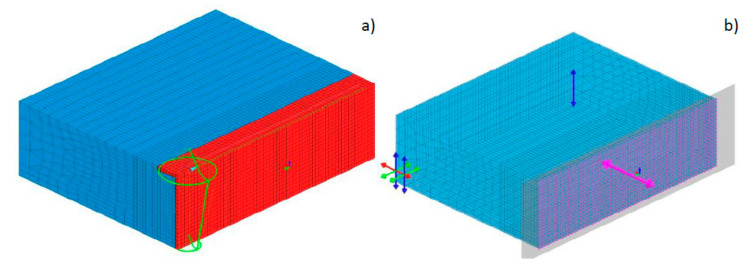
View (**a**) of the symmetrical three-dimensional model of the laser bead-on-plate welding process together with a conical model of the heat source and the “LOAD” area, and (**b**) boundary clamping conditions.

**Figure 12 materials-13-02653-f012:**
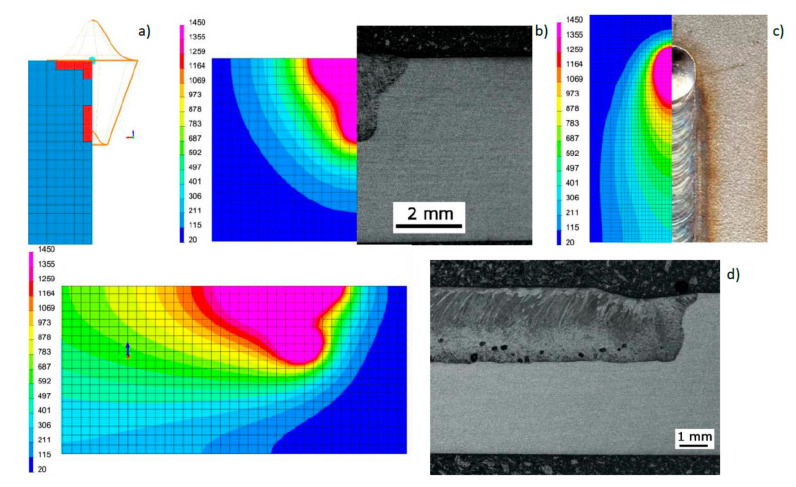
View (**a**) of the method of loading the mesh elements and placement of the conical model of the heat source, comparing (**b**) the shape of the bead on the cross-section, (**c**) the shape and size of the weld pool, and (**d**) the longitudinal section of DL_T1 test bead with the results of numerical analyses (Table 3 and Table 4).

**Figure 13 materials-13-02653-f013:**
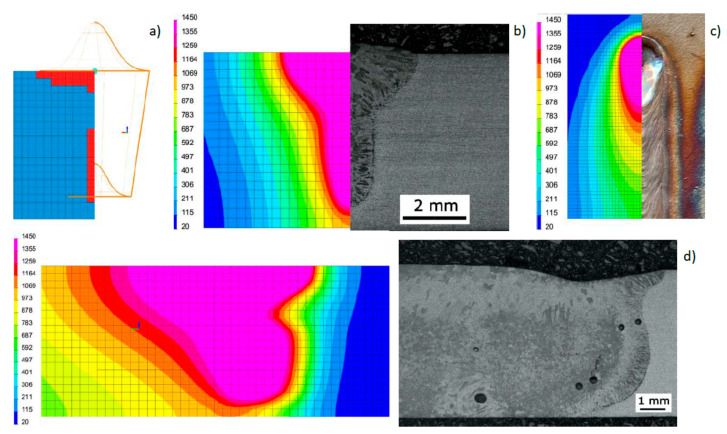
View (**a**) of the method of loading the mesh elements and placement of the conical model of the heat source, comparing (**b**) the shape of the bead on the cross-section, (**c**) the shape and size of the weld pool, and (**d**) the longitudinal section of DL_T3 test bead with the results of numerical analyses (Table 3 and Table 4).

**Figure 14 materials-13-02653-f014:**
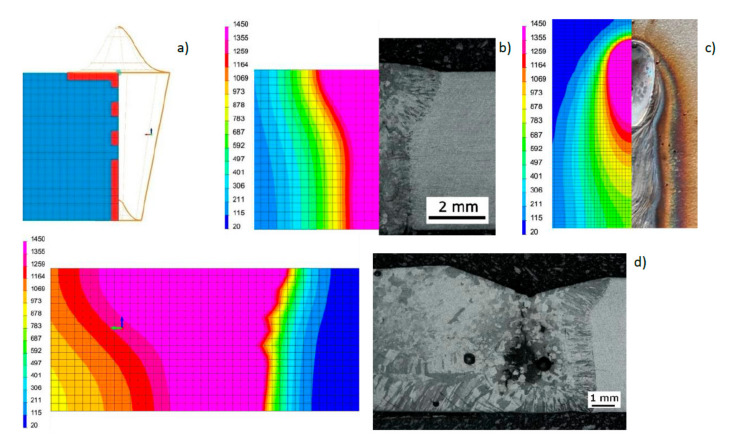
View (**a**) of the method of loading the mesh elements and placement of the conical model of the heat source, comparing **(b)** the shape of the bead on the cross-section, (**c**) the shape and size of the weld pool, and (**d**) the longitudinal section of DL_T5 test bead with the results of numerical analyses (Table 3 and Table 4).

**Figure 15 materials-13-02653-f015:**
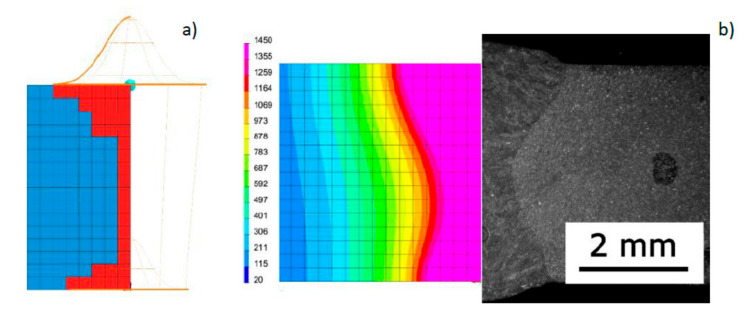
View (**a**) of the method of loading the mesh elements and placement of the conical model of the heat source, comparing (**b**) the shape of the DL1.5_8.33 bead on the cross-section with the results of numerical analysis (Table 1, Figure 7).

**Figure 16 materials-13-02653-f016:**
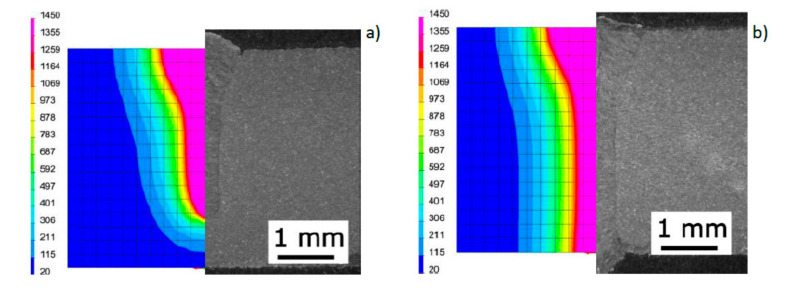
Comparison of the bead shape in the cross-section of sample (**a**) DL_TC3 and (**b**) DL_TC4 with the results of numerical analyses (Table 3).

**Figure 17 materials-13-02653-f017:**
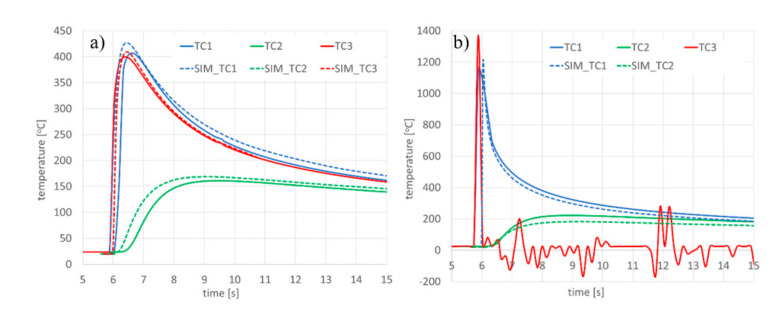
Comparison of recorded and calculated thermal cycles with the bead-on-plate laser welding process performed on 4.0-mm AISI 304 steel samples using the Trumpf TruDisk 3302 disk laser for sample (**a**) DL_TC3 and (**b**) DL_TC4. TCx: cycles recorded during the welding process; SIM_TCx: cycles obtained from numerical analyses.

**Figure 18 materials-13-02653-f018:**
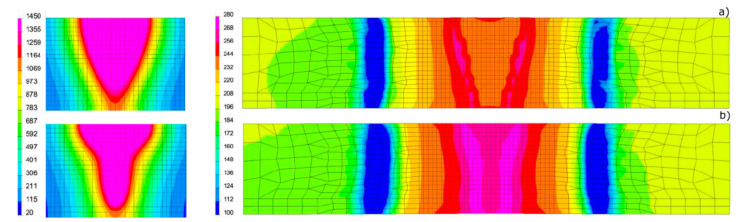
View of the molten pool and von Mises stresses distribution in the case of calculations for (**a**) the model without modification of the area of mesh elements loaded with the heat source model and (**b**) the model based on the proposed calculation modification for the DL_T3 sample (Table 3).

**Figure 19 materials-13-02653-f019:**
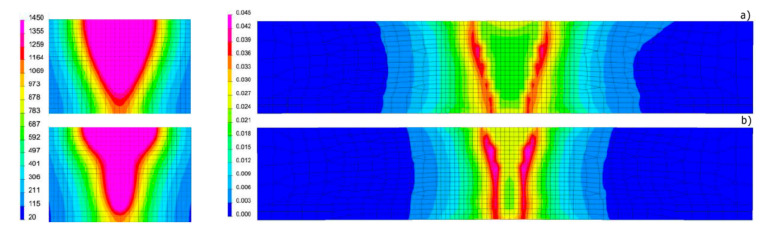
View of the molten pool and the distribution of cumulative plastic strains in the case of calculations for (**a**) the model without modification of the area of mesh elements loaded with the heat source and (**b**) the model developed based on the proposed modification for the DL_T3 sample (Table 3).

**Figure 20 materials-13-02653-f020:**
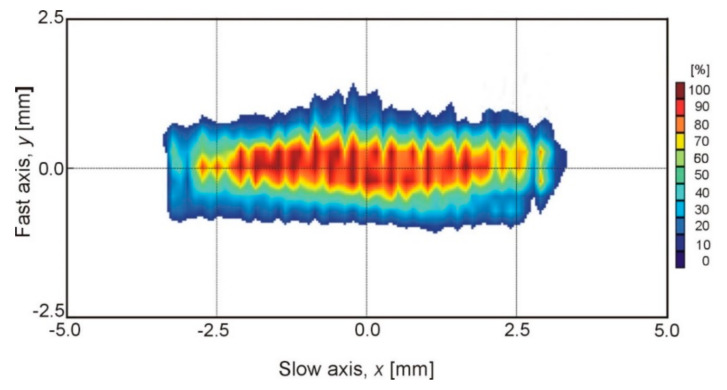
The 2D beam profile of the used HPDL laser in the focal plane [43]. (Reprinted with permission, copyright 2017, EBSCO Industries).

**Figure 21 materials-13-02653-f021:**
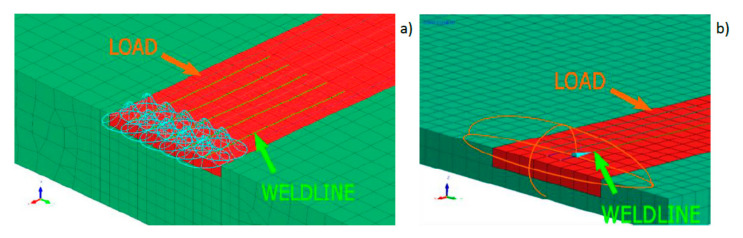
View of loading of the elements of the model mesh and placement of (**a**) heat source models with a normal distribution (Gaussian surface heat source model) and (**b**) a double-ellipsoid heat source model.

**Figure 22 materials-13-02653-f022:**
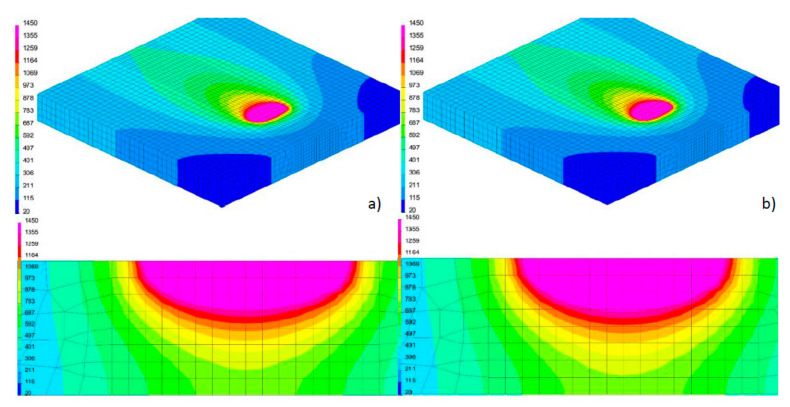
Comparison of the view of the molten pool surface and cross-sections (**a**) when using a model consisting of normal distributions and (**b**) a double-ellipsoid heat source model.

**Figure 23 materials-13-02653-f023:**
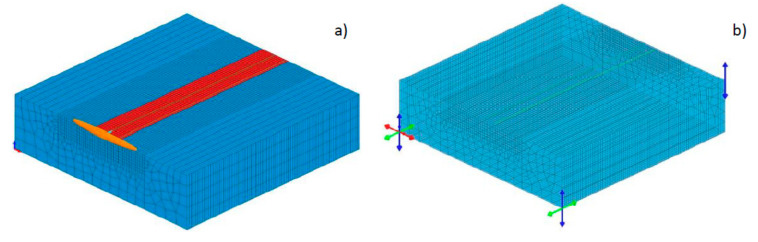
View (**a**) of a three-dimensional model of the bead-on-plate laser welding process performed on 6.0-mm steel samples using a high-power diode laser together with a double-ellipsoid heat source model and the loaded area (red color), and (**b**) clamping conditions.

**Figure 24 materials-13-02653-f024:**
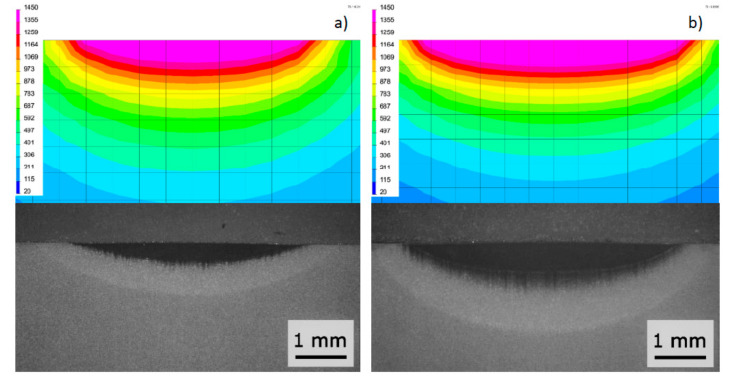
Comparison of the bead shape on the cross-section of sample (**a**) HPDL_R1 and (**b**) HPDL_R2 with the results of numerical analyses.

**Figure 25 materials-13-02653-f025:**
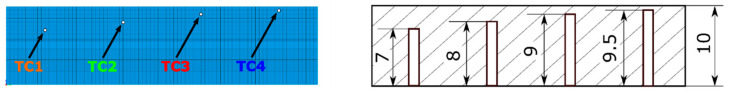
Scheme of thermocouple placement on the cross-section of samples in the axis of welded beads.

**Figure 26 materials-13-02653-f026:**
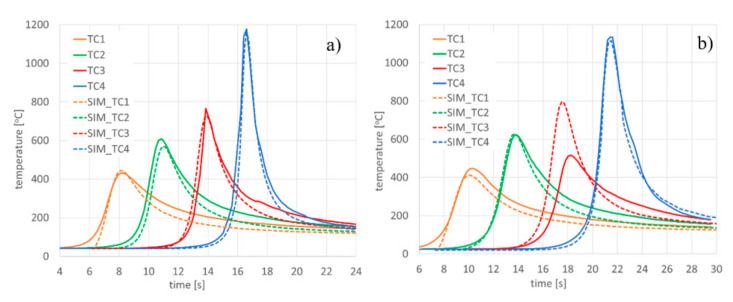
Comparison of recorded and calculated thermal cycles on the bead-on-plate laser welding process performed on 10-mm AISI304 steel samples using the ROFIN DL020 high-power diode laser with (**a**) TC3 and (**b**) TC4 samples. TCx: thermal cycles recorded during the welding process; SIM_TCx: thermal cycles obtained from numerical analyses.

**Table 1 materials-13-02653-t001:** Chemical composition of S355 (EN10025-2) and X5CrNi18-10 (EN10088-3) steel [40,41].

	Cr (%)	Ni (%)	Mn (%)	N (%)	Si (%)	C (%)	S (%)	P (%)
S355	-	-	1.5	-	0.55	0.20	0.035	0.035
AISI304	17.5–19.5	8.0–10.5	max. 2.0	max. 0.11	max. 0.75	max. 0.08	max. 0.03	max. 0.045

**Table 2 materials-13-02653-t002:** Comparison of geometric dimensions of beads in bead-on-plate welding tests performed on a 4.0-mm-thick sample of AISI304 steel using the Trumpf TruDisk 3302 disk laser with values obtained from numerical analysis (values in brackets) (Figure 7).

Specimen Designation	Bead Widthin the Face Area (mm)	Bead Width in the Root Area (mm)	Penetration Depth (mm)
DL1.0_8.33	2.44 (2.36)	1.12 (1.01)	3.16 (3.15)
DL1.5_8.33	3.04 (3.09)	2.3 (2.07)	4.0 (4.0)

**Table 3 materials-13-02653-t003:** Parameters of bead-on-plate laser welding of the 4.0- and 6.0-mm-thick AISI304 plates with the TRUMPF TruDisk 3302 laser.

Specimen Designation	Laser Beam Power (w)	Welding Speed (mm/s)	Energy Per Unit Length (J/mm)	Remarks
DL_T1	1000	8.33	120	Partial penetration, bell shape
DL_T2		16.6	60	
DL_T3	2000	8.33	240	
DL_T4		16.6	120	
DL_T5	3000	8.33	360	Full penetration, narrow root
DL_T6		16.6	180	
DL_TC_1	1400	16.6	84	Partial penetration, bell shape
DL_TC_3	1800	33.3	54	
DL_TC_2	1800	16.6	108	Full penetration, narrow root
DL_TC_4	2400	33.3	72	

Remarks—Argon flow rate via cylindrical nozzles: 15 L/min, laser spot diameter: 200 µm, location of the laser beam focus: on the upper surface of the welded sample. TC1_4 samples: 4.0-mm-thick samples equipped with sets of 3 thermocouples welded to the bottom of the sample according to Figure 10.

**Table 4 materials-13-02653-t004:** Comparison of geometric dimensions of selected beads on the bead-on-plate laser welding process performed on 6.0-mm AISI 304 steel samples using the Trumpf TruDisk 3302 disk laser with values obtained from numerical analysis (values in brackets) (Table 3, Figure 12, Figure 13 and Figure 14).

Specimen Designation	Bead Widthin the Face Area (mm)	Bead Width in the Root Area (mm)	Maximum Penetration Depth (mm)	Molten Pool Length (mm)
DL_T1	3.07 (3.18)	1.04 (0.96)	2.71 (2.71)	4.3 (4.6)
DL_T3	4.44 (4.46)	1.56 (1.44)	5.11 (5.33)	7.0 (7.08)
DL_T5	4.38 (4.37)	1.88 (1.79)	6.0 (6.0)	7.5 (7.54)

**Table 5 materials-13-02653-t005:** Comparison of geometric dimensions of selected beads on the bead-on-plate laser welding process performed on 6.0-mm AISI 304 steel samples using the ROFIN DL020 high-power diode laser with values obtained from numerical analyses (values in brackets) (Figure 21 and Figure 22).

Specimen Designation	Heat Source Model	Heat Source Dimension * (mm)	Energy Per Unit Length ** (J/mm)	Bead Width (mm)	Penetration Depth (mm)	Molten Pool Length (mm)
HPDL_1	Gaussian surface heat source model	1.0/1.0	45 for each of 12 models (total 560)	4.50	1.05	2.72
HPDL_2		1.0/0.5	25 for each of 22 models (total 560)	4.35	1.13	2.84
HPDL_3	Double ellipsoid model	15/1.8/0.1	560	5.01	1.01	2.74
HPDL_4		25/1.8/0.1		5.07	0.99	2.68
HPDL_5		25/1.8/0.5		4.98	1.00	2.66
HPDL_6		25/1.0/0.1		5.03	1.02	2.46
HPDL_7		50/1.8/0.1		5.09	0.97	2.66
HPDL_8		25/1.8/1.0		4.95	1.02	2.64

* Dimensions for the Gaussian surface heat source model: diameter/distance between trajectories (mm); for the double-ellipsoid model: width/length/height of model. ** Heat transfer efficiency coefficient: 0.5 (determined on the basis of model calibration based on actual bead dimensions).

**Table 6 materials-13-02653-t006:** Chemical composition of X37CrMoV5-1 steel according to ISO 4957:2018 standard [44].

	C	Mn	Si	P	S	Cr	Mo	V
WCL (X37CrMoV5-1)	0.32–0.42	0.2–0.5	0.8–1.2	max. 0.030	max. 0.030	4.5–5.5	1.2–1.5	0.3–0.5

**Table 7 materials-13-02653-t007:** Process parameters and comparison of geometric dimensions of beads made on 10 -mm WCL (X37CrMoV5-1) steel samples using a ROFIN DL020 high-power diode laser with values obtained from numerical analyses (values in brackets) (Figure 24).

Specimen Designation	Laser Beam Power (W)	Welding Speed (mm/s)	Energy Per Unit Length (J/mm)	Bead Width (mm)	Penetration Depth (mm)
*HPDL_R1*	1000	3.33	300	4.15 (4.10)	0.30 (0.35)
*HPDL_R2*	1250	2.5	500	5.18 (5.13)	0.50 (0.52)

Argon flow rate via cylindrical nozzle: 15 L/min; laser beam spot dimensions: 1.8 mm × 6.8 mm at 82-mm focal length; location of the laser beam focus: on the upper surface of the samples.

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
