# Peer review of "Heat Source Models in Numerical Simulations of Laser Welding"

_materials, 2020, doi:10.3390/ma13112653_

Round 1
Reviewer 1 Report
This paper presents a method to improve existing heat source models for finite element analysis of laser welding processes. Several examples are presented to describe the model calibration and validation. This research work should be of interest to the readers of Materials. I have some questions and suggestions:
- In Section 2.1, should the word “Furrier’s” appearing on Line 96 and Line 110 be modified to “Fourier’s”?
- The author should clarify what “axially symmetric” refers to specifically in Figure 11. Does the symmetry condition apply to the heat source model or the finite element model? Where is the axis of symmetry located? It looks like a plane of symmetry along the welding line is considered in the shown 3D finite element model, but this is different from an “axially symmetric” FE model.
- In Figure 17(b), why does the TC3 result look so different from TC1 and TC2? Why doesn’t the author plot the numerical result corresponding to TC3 in this figure?
- In Figure 26, the authors should discuss why the numerical model cannot capture the peak of TC4 well. What are the main sources of error?
- In the review of laser simulations, the following research work and references therein should be discussed, where advanced Gaussian-type and Cone-type hear source models are developed for moving side-by-side dual laser beams finite element modeling of laser welding processes with experimental validation:
He, Q., Wei, H., Chen, J. S., Wang, H. P., & Carlson, B. E. (2018). Analysis of hot cracking during lap joint laser welding processes using the melting state-based thermomechanical modeling approach. The International Journal of Advanced Manufacturing Technology, 94(9-12), 4373-4386.
Author Response
Thank you for the review of my article. Please find attached pdf file with the answers on all comments and suggestions.

Reviewer 2 Report
Generally speaking paper is interesting both from a scientific and an industrial point of view. Some improvements are suggested to the author as follows:
1. The entire text should be written in passive speech. Do not use e.g. "we have .." .. Page 1, rows 39-40 and so on.
2. Page 5, rows 190-193 for ratio 60:40, I suggest to smention the following reference where this ratio was applied (https://doi.org/10.1016/j.jmatprotec.2006.10.013)
3. Page 8, Table 1, Chemical composition ... give a reference where this information comes from? Same for Table 6.
4. Page 11, ..thermocouples (Ni-NiCr 0.2 mm). Are the thermocouples' diameter really 0.2 mm? What is the accuracy level of thermocouples?
5. Page 12, rows 394-400 ... the authors conducted a numerical simulation with three-dimensional elements. Provide more information about the finite elements: number of nodes, degrees of freedom, shape .... Furthermore, the thermal properties of the materials used are not visible anywhere. Density, thermal conductivity, specific thermal capacity? Are the thermal properties taken as constants or are they temperature dependent? What is the influence of the choice of thermal properties of the material (temperature dependent or independent) on the heat affected zone or on the simulated temperature histories?
6. The paper does not describe the boundary conditions on the external surfaces of the welded model.
Are convection heat transfer coefficient and emmisivity taken as constant values ​​(https://doi.org/10.1002/er.4506) or as a function of temperature (https://doi.org/10.1016/j.marstruc.2010.05.002)?
7. Discussion of results and comments of ifigures should be deeper. Eg. Fig. 26...it is necessary to comment more widely on the deviation of numerical results and experimental measurements. Eg. green curves, peeks are equal but cooling curves are not equal. Could this be attributed to the simplified definition of external boundary conditions, convection heat transfer coefficient and emmisivity (https://doi.org/10.1002/er.4506), if not, what is the possible reason for the deviation?
8. The quality of figures 17 and 26 is unacceptable. Curves are too thin, letters and numbers are too small and poorly visible. New figures need to be made.
Author Response

(The authors gave the same response as above.)
